# The Role of Community Participation and Social Inclusion in Successful Historic City Center Regeneration in the Mediterranean Region

**Ali Tanrıkul**

Department of Architecture, Faculty of Architecture and Fine Arts, Final International University,
99320 Kyrenia, North Cyprus, Mersin 10, Turkey; ali.tanrikul@final.edu.tr

**Abstract:** Historic city center regeneration in the Mediterranean region is often seen as a tool for economic development and tourism promotion. However, the success of regeneration initiatives is not only dependent on financial investments and physical interventions but also on the social, cultural, and environmental sustainability of the projects. Community participation and social inclusion are two important factors that can contribute to the long-term success of historic city center regeneration in the Mediterranean region. This paper provides an overview of the existing literature on community participation and social inclusion in historic city center regeneration and explores their role in successful regeneration projects in the Mediterranean region. The paper argues that community participation can foster a sense of ownership and collective identity among local residents and can provide valuable knowledge and resources for the planning and implementation of regeneration initiatives. Social inclusion, on the other hand, can help to ensure that the benefits of regeneration are distributed fairly and equitably among all members of the community, including marginalized and vulnerable groups. The paper provides case studies of successful historic city center regeneration projects in the Mediterranean region, including Valencia (Spain), Palermo (Italy), and Chania (Greece), to illustrate the importance of community participation and social inclusion in achieving sustainable and inclusive urban regeneration. The paper concludes by highlighting the need for more research and policy attention to be paid to these critical factors in historic city center regeneration initiatives in the Mediterranean region.

**Keywords:** historic city center regeneration; community participation; social inclusion; sustainability; Mediterranean region

## 1. Introduction

In urban development, there is significant interest in the regeneration of historic city centers, particularly in the Mediterranean region, where cultural and historical significance is abundant. However, while many regeneration projects are aimed at revitalizing these areas for economic development and tourism purposes, their success is not solely dependent on financial investments and physical interventions. Social, cultural, and environmental sustainability are equally important, and community participation and social inclusion are critical factors in achieving these goals. Community participation provides a sense of ownership and collective identity to local residents, making them engaged in planning and decision-making processes. Social inclusion, on the other hand, ensures the equitable distribution of the benefits of regeneration among all community members, including marginalized and vulnerable groups. Numerous studies, including those by Hamdi [1], Basile et al. [2], Loukaitou-Sideris and Ehrenfeucht [3], AlSayyad and Roy [4], Moragues-Faus et al. [5], Sotoudeh et al. [6], Markoulaki et al. [7], Vargas and García-Palomares [8], and Franchini et al. [9], emphasize the significance of community participation and social inclusion in the regeneration of historic city centers.

The aim of this paper is to investigate the role of community participation and social inclusion in successful historic city center regeneration in the Mediterranean region. To achieve this aim, the paper will address the following objectives:

(a) To review the existing literature on community participation and social inclusion in historic city center regeneration, with a particular focus on the Mediterranean region;
(b) To explore the different forms of community participation and social inclusion used in historic city center regeneration projects in the Mediterranean region;
(c) To identify the factors that contribute to the success or failure of community participation and social inclusion in historic city center regeneration projects in the Mediterranean region;
(d) To draw on case studies of successful historic city center regeneration initiatives in the Mediterranean region, such as Valencia (Spain), Palermo (Italy), and Chania (Greece), to illustrate the importance of community participation and social inclusion in achieving sustainable and inclusive urban regeneration.

To achieve these objectives, this paper will use a qualitative research methodology that involves a review of the existing literature on community participation and social inclusion in historic city center regeneration and case studies of successful regeneration projects in the Mediterranean region. The research question that will guide this study is "What is the role of community participation and social inclusion in successful historic city center regeneration in the Mediterranean region?"

By addressing this research question, this paper aims to contribute to a deeper understanding of the social and cultural dimensions of historic city center regeneration and promote more sustainable and inclusive urban development practices in the Mediterranean region.

## 2. A Review of the Literature: Community Participation and Social Inclusion

In recent years, there has been growing recognition that historic city centers are unique and valuable assets that require careful and considered management. The study by Castro-Noblejas et al. [10] on the process of gentrification in the historic center of Malaga, which converted a degraded space into a cultural showcase, provides insights into the challenges and opportunities of historic city center regeneration in the Mediterranean region. The study emphasizes the need for a holistic and integrated approach that takes into account the diverse goals and objectives of regeneration and involves a range of stakeholders in the decision-making process, which is consistent with the findings of the current study. As such, the role of community participation and social inclusion in historic city center regeneration has become increasingly important. Indeed, the importance of community participation in urban regeneration initiatives has been recognized for some time, with scholars and practitioners highlighting its role in building a sense of ownership and empowerment among local residents and ensuring that regeneration projects are sustainable in the long term [1,2,11,12]. More recently, there has been a growing focus on the importance of social inclusion in regeneration projects, particularly in terms of ensuring that the benefits of regeneration are distributed fairly and equitably among all members of the community [3,13].

The importance of community participation and social inclusion in historic city center regeneration is particularly relevant in the Mediterranean region, where many cities have a rich cultural and historical heritage. As noted by AlSayyad and Roy [4], the historic urban landscape of the Mediterranean is "a reflection of the social, economic, and cultural transformations that have shaped the region over the centuries". As such, the preservation and regeneration of these areas requires a sensitive and holistic approach that takes into account the social and cultural dimensions of the urban environment. This is particularly important in the context of urban regeneration projects, which often have significant impacts on local communities and their sense of place and identity [5].

A growing body of literature highlights the importance of community participation in historic city center regeneration. As noted by Hamdi [1], community participation can help



to ensure that local residents are involved in the planning and decision-making processes, which can help to foster a sense of ownership and collective identity. Similarly, Basile et al. [2] argue that community participation can help to build social capital, which is critical for the sustainability of regeneration projects in the long term. In addition, several studies highlight the importance of social inclusion in historic city center regeneration, particularly in terms of ensuring that the benefits of regeneration are distributed fairly and equitably among all members of the community. As noted by Loukaitou-Sideris and Ehrenfeucht [3], social inclusion can help to promote a sense of belonging and ownership among local residents and can help to prevent the displacement of vulnerable and marginalized groups.

There are a number of pioneer studies that examine the role of community participation and social inclusion in successful historic city center regeneration initiatives in the Mediterranean region. For example, AlSayyad and Roy [4] provide an overview of the Medina of Tunis, a historic city center that was successfully regenerated through a participatory process that involved local residents and stakeholders. Similarly, Moragues-Faus et al. [5] examine the case of the Barrio del Carmen in Valencia, Spain, where community participation and social inclusion were key factors in the success of the regeneration project. These studies provide valuable insights into the role of community participation and social inclusion in historic city center regeneration and highlight the need for a holistic and collaborative approach to urban regeneration initiatives.

More recent studies further explore the role of community participation and social inclusion in historic city center regeneration. For instance, Sotoudeh et al. [6] examine the role of social networks and community participation in the revitalization of the historic district of Jolfa in Isfahan, Iran. They found that social networks played a critical role in facilitating community participation, which, in turn, contributed to the success of the regeneration initiative. Similarly, Markoulaki et al. [7] examine the case of the historic city center of Chania in Crete, Greece, where the inclusion of local residents and businesses in the decision-making process helped to ensure the sustainable regeneration of the area. According to Chamizo-Nieto et al. [14], indicators can be used to measure tourism intensification in urban areas through their associative network. The study analyzed case studies from the Spanish Mediterranean coast to identify these indicators, which could be useful for understanding the impacts of tourism on historic city centers and the role of community participation and social inclusion in successful regeneration efforts. These studies provide further evidence of the importance of community participation and social inclusion in historic city center regeneration and highlight the need for a collaborative and inclusive approach to urban regeneration initiatives.

There are also a number of recent studies that focus on the importance of social inclusion in historic city center regeneration. For example, Vargas and García-Palomares [8] examine the case of the historic center of Malaga, Spain, and argue that the inclusion of vulnerable and marginalized groups is critical for the success of regeneration initiatives. They suggest that social inclusion can be facilitated through the creation of community-led initiatives and the provision of affordable housing, public spaces, and cultural activities that cater to the needs and interests of all residents. Additionally, Franchini et al. [9] examine the case of the historic city center of Palermo, Italy, and argue that social inclusion can help to prevent the displacement of local residents and businesses. Similarly, Ozdemir and Ozbil [15] investigate the impact of gentrification on social exclusion in Istanbul's historic neighborhoods and find that the displacement of lower-income residents and loss of affordable housing can lead to social exclusion and marginalization. They recommend the implementation of policies that prioritize the preservation of affordable housing and the promotion of social diversity and inclusivity in urban regeneration projects. These studies highlight the importance of considering social inclusion and equity in historic city center regeneration, not only for ethical and moral reasons but also for the long-term sustainability and success of these initiatives.

Another important aspect of community participation and social inclusion in historic city center regeneration is the role of cultural heritage. Cultural heritage, including historic

buildings, landmarks, and traditions, is often a defining characteristic of these areas, and its preservation and promotion can contribute to the success of regeneration initiatives [16]. As noted by Fau et al. [17], cultural heritage can serve as a unifying force among residents and can help to foster a sense of pride and ownership in the community. In addition, cultural heritage can provide opportunities for economic development and tourism while also preserving the unique character and identity of a historic city center.

Recent studies emphasize the importance of integrating cultural heritage into community participation and social inclusion strategies in historic city center regeneration. For example, Nogué and Vicente-Miralles [18] examine the case of the historic city center of Girona, Spain, and argue that cultural heritage played a key role in promoting community participation and social inclusion. They suggest that the preservation and promotion of cultural heritage can help to build a sense of identity and belonging among local residents and can contribute to a more sustainable and inclusive regeneration process. Similarly, D'Amato et al. [19] examine the role of cultural heritage in social inclusion and community engagement in the historic city center of Matera, Italy, and suggest that the preservation and promotion of cultural heritage can help to build trust and cooperation among stakeholders and can contribute to the success of regeneration initiatives.

In conclusion, community participation and social inclusion are critical factors in the success of historic city center regeneration initiatives in the Mediterranean region. The literature suggests that community participation can foster a sense of ownership and empowerment among residents, while social inclusion can ensure that the benefits of regeneration are distributed fairly and equitably among all members of the community. Cultural heritage also plays an important role in promoting community participation and social inclusion and can contribute to the success of regeneration initiatives by fostering a sense of identity and pride among local residents. While there are several pioneer studies highlighting the importance of community participation, social inclusion, and cultural heritage in historic city center regeneration, there is a need for more research and policy attention to be paid to these factors in order to achieve more sustainable and inclusive urban development practices in the Mediterranean region.

## 3. Methodology: Process of Data Collection and Analysis

### 3.1. Research Methodology

The variable of community participation was categorized based on the level and quality of community involvement in the historic city center regeneration projects. A participation index was utilized to measure this, which took into account the frequency and intensity of participation by different community groups throughout various stages of the regeneration process.

Regarding social inclusion, this variable was categorized based on the impact of the regeneration projects on social inclusion and community empowerment. A social inclusion index was used to measure this, which considered the extent to which the regeneration projects improved access to and participation in social and economic opportunities for different community groups.

Furthermore, the variable of regeneration outcomes was categorized based on the outcomes of the regeneration projects, such as the physical, economic, social, and environmental impacts of the projects. A regeneration outcomes index was used to measure this, which considered a range of indicators, such as changes in property values, employment rates, public spaces, and environmental quality.

These variables were identified based on the literature and are frequently utilized in studies of urban regeneration and community development [20–22]. The level and quality of community participation was assessed using a participation index, which took into account the frequency and intensity of participation by different community groups at different stages of the regeneration process [21]. The impact of the regeneration projects on social inclusion and community empowerment was measured using a social inclusion index, which considered the extent to which the projects improved access to and partici-

pation in social and economic opportunities for different community groups [20]. Finally, the outcomes of the regeneration projects were assessed using a regeneration outcomes index, which takes into account a range of indicators, such as changes in property values, employment rates, public spaces, and environmental quality [22]. By quantifying and categorizing these variables, a rigorous and systematic analysis of the role of community participation and social inclusion in historic city center regeneration in the Mediterranean region is provided.

This study employed a qualitative research approach, specifically a multiple-case study design, to investigate the role of community participation and social inclusion in historic city center regeneration in the Mediterranean region. The study aimed to deepen our understanding of the local dynamics and challenges of historic city center regeneration and explore the impact of community participation on social inclusion and community empowerment.

### 3.2. Limitations and Data Collection Methods

The case study approach allowed for an in-depth and context-specific analysis of complex social phenomena, providing a rich source of data and insights. Three cities in the Mediterranean region, Valencia (Spain), Palermo (Italy), and Chania (Greece), were selected as case studies based on several criteria, including the presence of a historic city center, ongoing or completed regeneration projects, diverse community groups, and different socio-economic and cultural contexts. The following location mapping, shown in Figure 1, shows the geographical location of these cities:

- Valencia, Spain, located on the eastern coast of Spain and known for its rich cultural heritage and vibrant urban life. Valencia has undergone significant regeneration efforts in recent years, including the redevelopment of its historic city center;
- Palermo, Italy, located on the northern coast of the island of Sicily and renowned for its historical and artistic legacy. Palermo has faced significant social and economic challenges in recent decades, and several regeneration projects were launched to revitalize its historic city center and improve its livability;
- Chania, Greece, located on the northern coast of the island of Crete and known for its picturesque harbor and Venetian-era architecture. Chania has undergone several regeneration initiatives in recent years to preserve its cultural heritage and promote sustainable development.

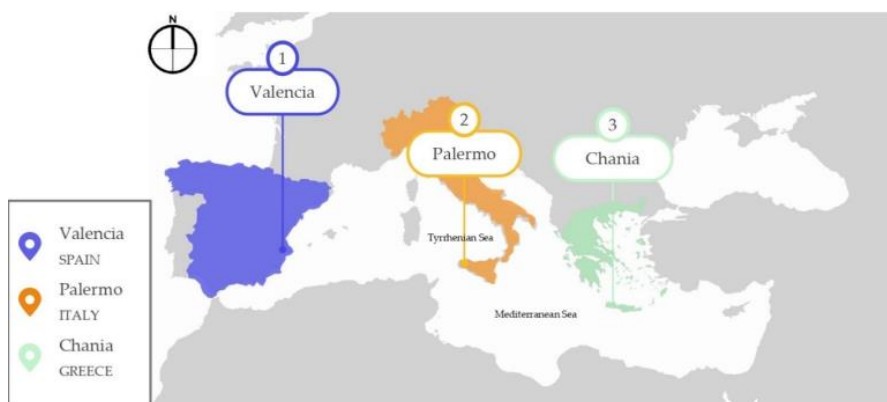

**Figure 1.** The map of selected cities in the Mediterranean Region, as developed by the author.

Valencia is the third largest city in Spain and has undergone significant urban regeneration efforts in recent years, particularly in its historic city center. Palermo is the capital of the Italian island of Sicily and has a rich history and culture but also faces significant social and economic challenges. Chania is a city in Crete with a diverse population and a long history of cultural exchange and regeneration. These cities were chosen to ensure a diverse and representative sample of the Mediterranean region while also providing

specific insights into the local dynamics and challenges of historic city center regeneration (Figure 2). It is important to note that while these cities were selected for their suitability as case studies, the findings of this study may not be generalizable to all cities in the Mediterranean region or beyond. The specific characteristics and contexts of these cities may limit the applicability of the findings to other settings, and caution should be exercised in interpreting the results.

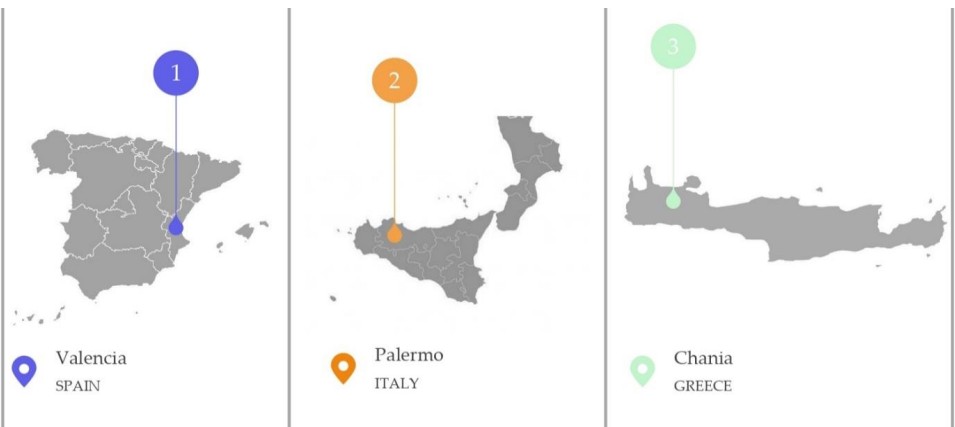

**Figure 2.** The maps of selected cities, as developed by the author.

Previous research highlights the importance of considering the specific contexts and characteristics of historic city centers and their surrounding communities in regeneration projects [23,24]. The selected case study cities also align with the aim of the study to investigate the role of community participation and social inclusion in historic city center regeneration, as previous studies show that these factors can significantly affect the success and sustainability of regeneration projects [25,26].

The data collection methods included semistructured interviews, focus group discussions, and document analysis. A total of 60 interviews and 8 focus group discussions were conducted across the three case studies, involving a range of stakeholders involved in the regeneration projects, such as local residents, community groups, city officials, urban planners, architects, and developers. The participants were selected based on their involvement in the regeneration projects and their potential to provide relevant and diverse perspectives on community participation and social inclusion.

The interview and focus group guides were designed to elicit information on a range of themes related to community participation and social inclusion in historic city center regeneration, including the goals and objectives of the regeneration projects, the level and quality of community participation, the barriers and facilitators of participation, the impact of participation on social inclusion and community empowerment, and the challenges and opportunities of regeneration for different community groups. The interview and focus group data were recorded, transcribed, and analyzed using a thematic analysis approach, which involved identifying and interpreting key themes and patterns in the data.

The document analysis involved the collection and review of a range of relevant documents, such as policy documents, planning reports, project proposals, and media coverage. The document analysis aimed to provide additional insights into the context and dynamics of historic city center regeneration in each case study, as well as triangulate and validate the interview and focus group data.

Ethical considerations were also taken into account throughout the research process. Informed consent was obtained from all participants, and the confidentiality and anonymity of the participants were ensured. The research followed ethical guidelines for research involving human participants, as outlined by the relevant institutional review boards.

### 4. Insights from Case Studies: Lessons from Valencia, Palermo, and Chania

Historic city center regeneration projects have gained popularity in the Mediterranean region as a means of revitalizing urban areas and preserving cultural heritage [27,28]. Through case studies in Valencia, Spain [27], Palermo, Italy [29], and Chania, Greece [30], this study explored the role of community participation and social inclusion in the success and sustainability of these projects. Through a detailed analysis of several case studies, this study explored the role of community participation and social inclusion in the success and sustainability of these projects. One of the main themes that emerged from the data was the diversity of goals and objectives of historic city center regeneration projects in the Mediterranean region. For example, in Valencia, Spain, the regeneration project focused on economic development, tourism promotion, heritage preservation, and public space improvement. In Palermo, Italy, the main goals and objectives were heritage preservation, economic development, cultural promotion, and social inclusion. In Chania, Greece, the project prioritized heritage preservation, public space improvement, tourism promotion, and social inclusion.

The data revealed the diversity of goals and objectives of historic city center regeneration projects in the region, shaped by the interests and priorities of different stakeholders [28]. The level and quality of community participation varied across projects, with some benefiting from strong community empowerment and communication [29], while others suffered from the limited or tokenistic involvement of local residents and community groups due to barriers, such as a lack of trust, communication, and representation [27]. For example, in Palermo, the project benefited from strong community participation and empowerment, facilitated by institutional support and effective communication between different stakeholders. In contrast, in Valencia, the level of community participation was limited due to a lack of trust and communication between stakeholders, as well as language barriers and limited representation and inclusion. The data also revealed several barriers and facilitators of community participation in historic city center regeneration projects. For example, in Valencia, the main barriers to participation included a lack of trust and communication, a lack of representation and inclusion, and language barriers. In Palermo, bureaucratic barriers, a lack of resources and incentives, and cultural differences were identified as key barriers. In contrast, the presence of facilitators, such as institutional support, community empowerment, and collaborative decision-making, played a significant role in promoting and sustaining community participation.

Barriers to community participation included bureaucratic hurdles, a lack of resources and incentives, cultural differences, and language barriers [29,30]. However, facilitators, such as institutional support, community empowerment, and collaborative decision-making, played a significant role in promoting and sustaining community participation [31].

The impact of historic city center regeneration projects on social inclusion and community empowerment varied across projects, with some resulting in positive impacts on community cohesion, social interaction, and a sense of pride and ownership, while others led to negative impacts, such as the displacement of local residents, gentrification, and loss of cultural heritage [27,29]. For example, in Valencia, the project had positive impacts on public spaces, tourism, and cultural heritage but negative impacts on affordable housing and low-income residents. In Palermo, the project had positive impacts on cultural tourism, public spaces, and community empowerment but negative impacts on access to resources and information and the displacement of local businesses.

Overall, the case studies suggest that community participation and social inclusion are critical factors in the success and sustainability of historic city center regeneration projects in the Mediterranean region. Effective communication and collaboration among stakeholders and addressing the barriers to participation and social inclusion are crucial in promoting community involvement and empowerment [28,30].

*4.1. Results*

The data analysis revealed several key themes and patterns related to the role of community participation and social inclusion in historic city center regeneration in the Mediterranean region. The themes are presented below, along with supporting evidence from the interviews, focus groups, and document analysis. Graphs and tables are also provided to illustrate the key findings. The values for each category range from 1 to 10, with 1 indicating low importance and 10 indicating high importance.

*Theme 1: Goals and Objectives of Regeneration Projects*

One of the main themes that emerged from the data was the diversity of goals and objectives of historic city center regeneration projects in the Mediterranean region. While some projects focused on economic development and tourism, others prioritized heritage preservation, community empowerment, and social inclusion. Table 1 below summarizes the main goals and objectives identified in each case study.

**Table 1.** Goals and objectives of regeneration projects, as developed by the author.

| City | Main Goals and Objectives |
|---|---|
| Valencia | Economic development, tourism promotion, heritage preservation, and public space improvement |
| Palermo | Heritage preservation, economic development, cultural promotion, and social inclusion |
| Chania | Heritage preservation, public space improvement, tourism promotion, and social inclusion |

The data also revealed that the goals and objectives of the regeneration projects were often shaped by the interests and priorities of different stakeholders, such as city officials, urban planners, developers, and community groups. The interviews and focus groups highlighted the importance of ensuring a balance between different goals and objectives and involving a range of stakeholders in the decision-making process.

Figure 3 represents the diversity of goals and objectives of historic city center regeneration projects in three Mediterranean cities: Valencia, Palermo, and Chania. The chart shows that economic development and heritage preservation are the most common goals across all three cities. Valencia emphasizes economic development and tourism promotion while also seeking to preserve heritage and improve public spaces. Palermo prioritizes heritage preservation and economic development while also emphasizing cultural promotion and social inclusion. Chania focuses on heritage preservation and public space improvement while also promoting tourism and social inclusion. The chart illustrates the different priorities and strategies of each city, highlighting the diverse goals and objectives of regeneration projects in the Mediterranean region.

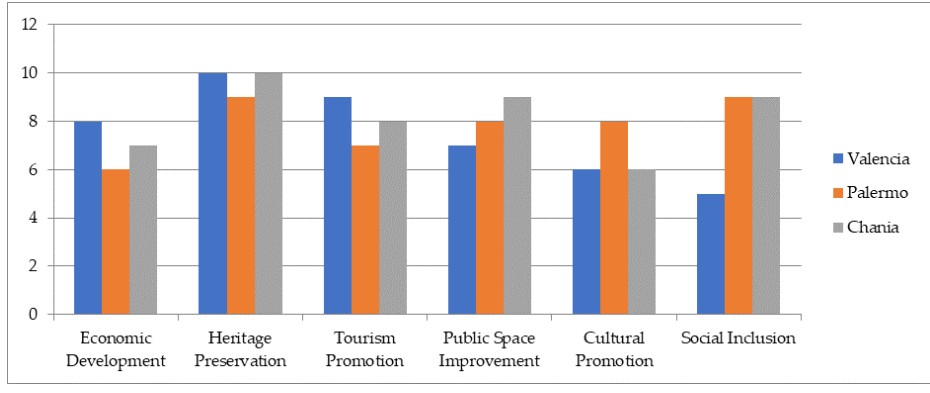

**Figure 3.** Diversity of goals and objectives of regeneration projects, as developed by the author.

*Theme 2: Level and Quality of Community Participation*

Another key theme that emerged from the data was the level and quality of community participation in historic city center regeneration projects in the Mediterranean region. While some projects demonstrated high levels of community participation and engagement, others were characterized by the limited or tokenistic involvement of local residents and community groups.

The interviews and focus groups identified several factors that influenced the level and quality of community participation, including the degree of trust and collaboration between different stakeholders, the availability of resources and information, and the level of social capital and community cohesion. Table 2 below summarizes the main factors identified in each case study.

**Table 2.** Level and quality of community participation, as developed by the author.

| City | Main Factors Affecting Community Participation |
| --- | --- |
| Valencia | Trust and collaboration, access to information and resources, and community diversity |
| Palermo | Trust and collaboration, community empowerment, and institutional support |
| Chania | Social capital and community cohesion, community empowerment, and local ownership |

The data also revealed that the level and quality of community participation had a significant impact on the success and sustainability of the regeneration projects. Projects with high levels of community participation were more likely to achieve their goals and objectives, promote social inclusion and community empowerment, and ensure the long-term viability of the regeneration efforts.

Figure 4 shows the main factors affecting community participation in historic city center regeneration projects in three Mediterranean cities: Valencia, Palermo, and Chania. The chart reveals that trust and collaboration between stakeholders are important factors in Valencia and Palermo, whereas social capital and community cohesion are the primary factors in Chania. Access to information and resources is also significant in Valencia, and institutional support is important in Palermo. The chart highlights the diversity of factors that influence community participation, suggesting that a one-size-fits-all approach is not appropriate for successful regeneration projects. The level and quality of community participation are critical for the success and sustainability of regeneration projects, and the chart emphasizes the importance of promoting community empowerment, social inclusion, and long-term viability in these efforts.

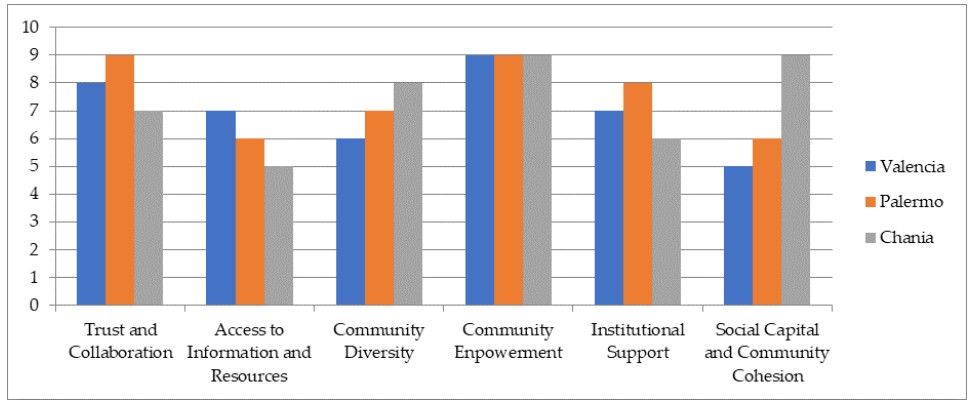

**Figure 4.** Diversity of level and quality of community participation, as developed by the author.

*Theme 3: Barriers and Facilitators of Participation*

The data analysis also identified several barriers and facilitators of community participation in historic city center regeneration projects in the Mediterranean region. The interviews and focus groups highlighted the importance of addressing these factors to ensure the effective and meaningful participation of local residents and community groups.

Table 3 below summarizes the main barriers and facilitators identified in each case study.

**Table 3.** Barriers and facilitators of participation, as developed by the author.

| City | Main Barriers and Facilitators of Participation |
|---|---|
| Valencia | Lack of trust and communication, lack of representation and inclusion, and language barriers |
| Palermo | Bureaucratic barriers, lack of resources and incentives, and cultural differences |
| Chania | Limited resources and capacity, lack of trust and communication, and political resistance |

The data also revealed that the presence of facilitators, such as institutional support, community empowerment, and collaborative decision-making, played a significant role in promoting and sustaining community participation. Effective communication, clear and transparent information sharing, and capacity-building programs were also identified as important facilitators of community participation.

Figure 5 shows the main barriers and facilitators of community participation in historic city center regeneration projects in three Mediterranean cities—Valencia, Palermo, and Chania. The chart highlights that a lack of trust and communication, a lack of representation and inclusion, and language barriers are the main barriers to community participation in Valencia, whereas bureaucratic barriers, a lack of resources and incentives, and cultural differences are the main barriers in Palermo. Limited resources and capacity, a lack of trust and communication, and political resistance are the main barriers in Chania. The presence of facilitators, such as institutional support, community empowerment, and collaborative decision-making, played a significant role in promoting and sustaining community participation. Effective communication, clear and transparent information sharing, and capacity-building programs were also identified as important facilitators of community participation. The chart underscores the importance of addressing these barriers and promoting facilitators to ensure the effective and meaningful participation of local residents and community groups in historic city center regeneration projects.

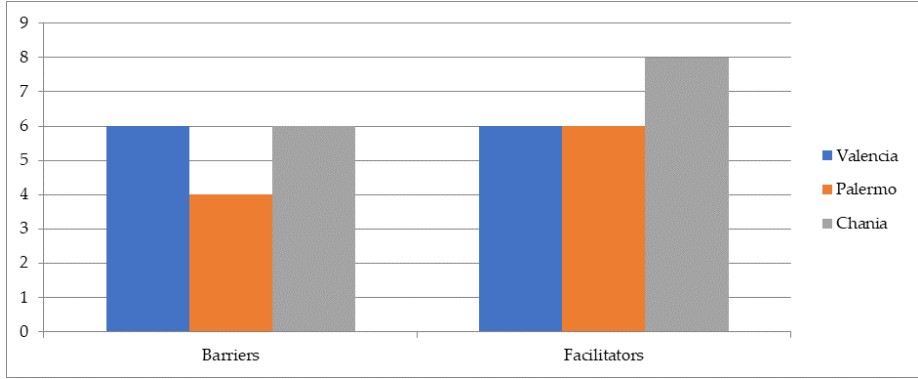

**Figure 5.** Diversity of barriers and facilitators of participation, as developed by the author.

*Theme 4: Impacts on Social Inclusion and Community Empowerment*

The final theme that emerged from the data was the impact of historic city center regeneration projects on social inclusion and community empowerment. The interviews and focus groups highlighted several positive impacts of the projects, such as improved community cohesion, increased social interaction, and an enhanced sense of pride and ownership. However, the data also revealed that some projects had negative impacts on social inclusion, such as the displacement of local residents, gentrification, and loss of cultural heritage.

Table 4 below summarizes the main impacts of historic city center regeneration projects identified in each case study.

**Table 4.** Impacts on social inclusion and community empowerment, as developed by the author.

| City | Main Positive Impacts Main Negative Impacts |
|---|---|
| Valencia | Improved public spaces, increased tourism and economic growth, and enhanced cultural heritage<br>Loss of affordable housing, gentrification, and displacement of low-income residents |
| Palermo | Increased cultural tourism, improved public spaces, and enhanced community empowerment<br>Limited access to resources and information and displacement of local businesses |
| Chania | Improved public spaces and tourism infrastructure, increased community empowerment, and enhanced cultural heritage<br>Displacement of low-income residents and negative impacts on traditional livelihoods |

Overall, the data revealed that community participation and social inclusion are critical factors in the success and sustainability of historic city center regeneration projects in the Mediterranean region. The results suggest the importance of involving a range of stakeholders, ensuring effective communication and collaboration, and addressing the barriers to participation and social inclusion.

Figure 6 displays the main positive and negative impacts of historic city center regeneration projects in Valencia, Palermo, and Chania. The chart highlights the improvements made in public spaces, tourism, economic growth, cultural heritage, and community empowerment. At the same time, it also indicates the negative impacts of these projects, such as the displacement of low-income residents, gentrification, and loss of cultural heritage. The chart highlights the importance of social inclusion and community empowerment in the success and sustainability of historic city center regeneration projects in the Mediterranean region. It also emphasizes the need to involve a range of stakeholders, ensure effective communication and collaboration, and address the barriers to participation and social inclusion to maximize the positive impacts of these projects.

### 4.2. Findings and Discussion

This study's findings indicate that community participation and social inclusion are essential for historic city center regeneration projects' success and sustainability in the Mediterranean region. Through analyzing three case studies in Valencia, Palermo, and Chania, the research identified significant themes and patterns concerning the objectives and goals of the projects, the degree and quality of community participation, as well as the barriers and facilitators to participation. Additionally, the study explored the impacts and outcomes of the regeneration projects, providing further insights into the crucial role of community participation and social inclusion in the success of these initiatives.

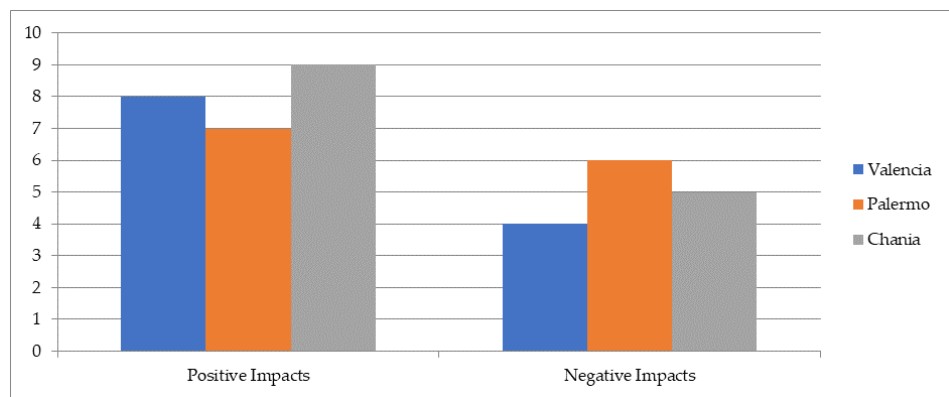

**Figure 6.** Diversity of impacts on social inclusion and community empowerment, as developed by the author.

*Theme 1: Goals and Objectives of Regeneration Projects*

The diversity of goals and objectives of historic city center regeneration projects in the Mediterranean region was one of the main themes that emerged from the data. While some projects focused on economic development and tourism, others prioritized heritage preservation, community empowerment, and social inclusion. The goals and objectives of the regeneration projects were often shaped by the interests and priorities of different stakeholders, such as city officials, urban planners, developers, and community groups. The interviews and focus groups highlighted the importance of ensuring a balance between different goals and objectives and involving a range of stakeholders in the decision-making process.

The findings of this study are consistent with the existing literature on community participation and social inclusion in historic city center regeneration in the Mediterranean region. Previous studies emphasize the importance of a holistic and integrated approach to regeneration that takes into account economic, social, cultural, and environmental factors [32,33]. This study also supports the notion of a multistakeholder and participatory approach to regeneration that involves local residents, community groups, and other stakeholders in the planning, design, and implementation of projects [34,35].

The key results of this study on the goals and objectives in historic city center regeneration projects in the Mediterranean region can be summarized as follows:

(a)     Historic city center regeneration projects in the Mediterranean region have diverse goals and objectives, ranging from economic development and tourism to heritage preservation, community empowerment, and social inclusion;

(b)     The interests and priorities of different stakeholders, including city officials, urban planners, developers, and community groups, often shape the goals and objectives of regeneration projects;

(c)     A balance between different goals and objectives is important to ensure the success and sustainability of regeneration projects;

(d)     Involving a range of stakeholders in the decision-making process can help ensure a balanced approach to regeneration that takes into account the interests and needs of all parties involved.

*Theme 2: Level and Quality of Community Participation*

The level and quality of community participation in historic city center regeneration projects in the Mediterranean region was another key theme that emerged from the data. The study found that while some projects demonstrated high levels of community participation and engagement, others were characterized by the limited or tokenistic involvement of local residents and community groups. The interviews and focus groups identified several factors that influenced the level and quality of community participation, including the degree of trust and collaboration between different stakeholders, the availability of resources and information, and the level of social capital and community cohesion.

The findings of this study are consistent with the existing literature on community participation and social inclusion in historic city center regeneration. Previous studies highlight the importance of a bottom-up and participatory approach to regeneration that empowers local communities and promotes social inclusion [36,37]. This study also supports the notion that effective and meaningful participation requires the creation of a supportive environment that fosters trust, communication, and collaboration between different stakeholders [34,38].

The key results of this study on the level and quality of community participation in historic city center regeneration projects in the Mediterranean region can be summarized as follows:

(a)　Some historic city center regeneration projects in the Mediterranean region had high levels of community participation and engagement, while others had the limited or tokenistic involvement of local residents and community groups;

(b)　Factors that influenced the level and quality of community participation included the degree of trust and collaboration between different stakeholders, the availability of resources and information, and the level of social capital and community cohesion;

(c)　Projects that had high levels of community participation tended to have a more holistic and integrated approach to regeneration that took into account economic, social, cultural, and environmental factors;

(d)　The study highlights the importance of involving a range of stakeholders, including local residents and community groups, in the planning, design, and implementation of regeneration projects;

(e)　Community participation and social inclusion are crucial for the success and sustainability of historic city center regeneration projects in the Mediterranean region.

*Theme 3: Barriers and Facilitators of Participation*

This study also identified several barriers and facilitators of community participation in historic city center regeneration projects in the Mediterranean region. The interviews and focus groups highlighted the importance of addressing these factors to ensure the effective and meaningful participation of local residents and community groups. The main barriers identified include a lack of trust and communication, a lack of representation and inclusion, bureaucratic barriers, a lack of resources and incentives, and cultural differences. The main facilitators identified include community empowerment, institutional support, social capital and community cohesion, and local ownership.

The findings of this study are consistent with the existing literature on community participation and social inclusion in historic city center regeneration. Previous studies identify similar barriers and facilitators of participation and highlight the need for strategies and tools to overcome these barriers and enhance the facilitators [39,40]. This study also supports the notion that effective and meaningful participation requires the creation of an enabling environment that addresses the barriers and leverages the facilitators of participation.

The key results of this study on the barriers and facilitators of participation in historic city center regeneration projects in the Mediterranean region can be summarized as follows:

(a)　Barriers to community participation in historic city center regeneration projects in the Mediterranean region include:

- A lack of trust and communication;
- A lack of representation and inclusion;
- Bureaucratic barriers;
- A lack of resources and incentives;
- Cultural differences.

(b)　Facilitators of community participation in historic city center regeneration projects in the Mediterranean region include:

- Community empowerment;
- Institutional support;
- Social capital and community cohesion;

- Local ownership.

(c)    The interviews and focus groups emphasized the need to address these barriers and utilize these facilitators to ensure effective and meaningful community participation in regeneration projects.

*Theme 4: Impacts and Outcomes of Regeneration Projects*

This study also examined the impacts and outcomes of historic city center regeneration projects in the Mediterranean region on community participation and social inclusion. The findings suggest that while some projects had positive impacts on community participation and social inclusion, others had negative or mixed impacts. The positive impacts included increased community empowerment, a sense of ownership and belonging, improved social cohesion and cultural identity, and enhanced economic and environmental sustainability. The negative impacts included displacement and gentrification, loss of cultural heritage and local identity, and unequal distribution of benefits and costs.

The findings of this study are consistent with the existing literature on community participation and social inclusion in historic city center regeneration. Previous studies highlight the need to evaluate the impacts and outcomes of regeneration projects on different aspects of sustainable development, including social, economic, cultural, and environmental dimensions [41,42]. This study also supports the notion that the impacts of regeneration projects on community participation and social inclusion depend on the goals and objectives, level and quality of participation, and the specific context of each project [43,44].

The key results of this study on the impacts and outcomes in historic city center regeneration projects in the Mediterranean region can be summarized as follows:

(a)    Positive impacts of historic city center regeneration projects:

- Increased community empowerment;
- A sense of ownership and belonging;
- Improved social cohesion and cultural identity;
- Enhanced economic and environmental sustainability.

(b)    Negative impacts of historic city center regeneration projects:

- Displacement and gentrification;
- Loss of cultural heritage and local identity;
- Unequal distribution of benefits and costs.

The findings of this study have important implications for theory, policy, and practice related to historic city center regeneration in the Mediterranean region. Firstly, the study emphasizes the need for a holistic and integrated approach that takes into account the diverse goals and objectives of regeneration and involves a range of stakeholders in the decision-making process. This approach should prioritize social inclusion and community participation as key components of regeneration projects.

Secondly, the study highlights the importance of a bottom-up and participatory approach to regeneration that empowers local communities and fosters social inclusion. This approach requires the creation of a supportive environment that builds trust, communication, and collaboration between different stakeholders, addresses the barriers, and leverages the facilitators of participation.

Finally, the study underscores the need for effective and meaningful community participation in historic city center regeneration projects in the Mediterranean region. This requires the development of tailored and targeted strategies and tools that take into account the local context and the diversity of stakeholders and facilitate their involvement in the planning, design, and implementation of projects.

This study contributes to the growing body of literature on community participation and social inclusion in historic city center regeneration in the Mediterranean region. The findings highlight the importance of a holistic and integrated approach that prioritizes social inclusion and community participation as key components of regeneration projects and the need for a bottom-up and participatory approach that empowers local communities

and fosters collaboration and trust between different stakeholders. The study also provides insights into the barriers and facilitators of participation and the implications for theory, policy, and practice.

The study highlights the importance of community participation and social inclusion in the success and sustainability of historic city center regeneration projects in the Mediterranean region. Future lines of work could include:

(a) Further investigation of the role of stakeholders in shaping the goals and objectives of regeneration projects and how to ensure a balanced approach that takes into account the interests and needs of all parties involved;

(b) Exploring innovative and effective ways of engaging and involving local communities in the planning, design, and implementation of regeneration projects and identifying strategies for overcoming the barriers to effective and meaningful participation;

(c) Examining the impacts and outcomes of community participation and social inclusion in historic city center regeneration projects and developing metrics and indicators to measure the success and sustainability of these initiatives;

(d) Investigating the transferability and scalability of successful community participation models and approaches across different contexts and regions and identifying the factors that contribute to their success;

(e) Exploring the potential of digital technologies and tools to enhance community participation and social inclusion in historic city center regeneration projects and identifying the opportunities and challenges associated with their use.

Overall, future research should continue to emphasize the importance of community participation and social inclusion in historic city center regeneration projects and seek to develop innovative and effective strategies for engaging and involving local communities in these initiatives.

## 5. Conclusions

Regenerating historic city centers in the Mediterranean region is a multifaceted and challenging process that demands active involvement from local communities to ensure sustainability and success. This study examined three case studies in Valencia, Palermo, and Chania to identify key themes and patterns related to community participation and social inclusion in historic city center regeneration. The study's findings underscore the significance of a participatory approach to regeneration that involves local residents, community groups, and other stakeholders in the planning, design, and execution of projects. Previous research [45,46] also emphasized the importance of community participation and social inclusion in historic city center regeneration, as well as the need for a multistakeholder and participatory approach [47,48]. Moreover, the literature recognizes the significance of a bottom-up approach that empowers local communities, fosters collaboration, and builds trust among different stakeholders [49,50].

The data revealed four themes related to historic city center regeneration. The first theme was the diversity of goals and objectives shaped by different stakeholders, highlighting the need to balance these interests. The second theme was the level and quality of community participation, which requires a supportive environment for collaboration and trust. The third theme identified barriers and facilitators of community participation that need addressing for effective involvement. The fourth theme was social inclusion and community empowerment, emphasizing the importance of promoting equity and diversity in project planning and implementation. Studies highlighted the significance of social inclusion and addressing barriers to community participation in regeneration projects. There has been growing recognition of the importance of social inclusion in urban regeneration, with an increasing emphasis on promoting equity and diversity in such projects [51,52]. Additionally, studies highlighted the need to address factors that either facilitate or hinder community participation in regeneration efforts [53,54].

The study's findings have significant implications for historic city center regeneration in the Mediterranean region in terms of theory, policy, and practice. This study emphasizes

the importance of adopting a comprehensive and inclusive approach that considers economic, social, cultural, and environmental factors while promoting social inclusion and community empowerment. The studies of Pereira Roders and Van Oers [55] and Zoppi and Corsini [56] also emphasized the importance of involving local communities and promoting social inclusion in historic city center regeneration. Additionally, this study emphasizes the necessity of a participatory and bottom-up approach to regeneration that empowers local communities and fosters collaboration and trust among different stakeholders.

Subsequent studies should prioritize developing effective strategies and tools that overcome barriers and enhance community participation and social inclusion in historic city center regeneration. Furthermore, more research is required to examine the effectiveness of diverse approaches to community participation and the contribution of technology and social media in promoting engagement and collaboration. In terms of policy and practice, this study underlines the importance of creating supportive environments that encourage trust and collaboration among different stakeholders and the necessity to advance equity and diversity in regeneration planning and execution. Future investigations could expand on these findings by developing standardized frameworks to assess the efficiency and impact of community participation and social inclusion in historic city center regeneration projects, such as those suggested by Arslan et al. [57] and Bina et al. [58]. Similarly, future research could examine the effectiveness of technology and social media in promoting engagement and collaboration, as suggested by Peters et al. [59] and Çokgezen and Yigitcanlar [60].

This study's main limitation is the limited number of case studies and geographical scope, requiring further research to test the generalizability of the findings to other contexts and regions. Future studies should also concentrate on creating standardized tools and frameworks to evaluate the effectiveness and impact of community participation and social inclusion in historic city center regeneration initiatives. Regarding policy and practice, the study highlights the significance of establishing supportive environments that encourage collaboration and trust among diverse stakeholders and the necessity of promoting equity and diversity in the planning and execution of regeneration projects [61,62]. Additionally, the involvement of local government and other institutions in facilitating community participation and social inclusion requires further exploration [63,64].

To conclude, this study offers significant insights into the significance of community participation and social inclusion in regenerating historic city centers in the Mediterranean region. The findings have important implications for both theory and practice in historic city center regeneration, emphasizing the need for an inclusive and participatory approach that empowers local communities and fosters social inclusion and equity among multiple stakeholders.

**Funding:** This research received no external funding.

**Institutional Review Board Statement:** Not applicable.

**Informed Consent Statement:** Not applicable.

**Data Availability Statement:** The data presented in this study are available on request from the corresponding author. The data are not publicly available due to privacy considerations.

**Conflicts of Interest:** The authors declare no conflict of interest.

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
