# Peer review of "The Role of Community Participation and Social Inclusion in Successful Historic City Center Regeneration in the Mediterranean Region"

_sustainability, doi:10.3390/su15097723_

Round 1
Reviewer 1 Report (Previous Reviewer 1)
The authors have sufficiently refined the original article. It can be published in its current form.
Author Response
Please see the attachment.

Reviewer 2 Report (Previous Reviewer 2)
Dear author,
I find your work very interesting and necessary for the scientific community and of great interest for society as there is a direct transfer to public policy.
However, I leave you some indications that can improve the work:
-Improve figure 1. You should elaborate a map with some more advanced Geographic Information System in which the basic elements of a map are included. North, scale, legend... In addition, it should include a small view at a less detailed scale to contextualise the study areas at a continental or global level.
-Section 4.2. I would call it "findings and discussion" since a good part of this task is developed in this section. In addition, I would insist on defining future lines of work derived from this interesting study.
-In addition, in order to contextualise the case of the city of Valencia and Mediterranean cities, the following works should be cited:
Castro-Noblejas, H. et al. (2022). Process of gentrification of a degraded space converted into a cultural showcase. El caso del Centro Histórico de Málaga. Revista de Estudios Andaluces, 43(43), Art. 43. https://doi.org/10.12795/rea.2022.i43.01
Chamizo-Nieto, F. J. et al. (2022). Indicators for measuring tourism intensification in urban areas through their associative network: Case studies from the Spanish Mediterranean coast. European Journal of Tourism Research, 32, 3202-3202. https://doi.org/10.54055/ejtr.v32i.2627
Without further ado, I congratulate you on your work.
Best regards.
Round 2
Reviewer 2 Report (Previous Reviewer 2)
Dear author/s,
The new references hasn't been added in the right way. The bibliography put other authors after the first author. It's essential correct that mistake in Chamizo's and in Castro's articles.
On the other hand, in the first map Greece doesn't appear like Italy and Spain, so it's necessary to correct this mistake.
I wait for the last version of your work to give a good visa.
Best regards.
Round 3
Reviewer 2 Report (Previous Reviewer 2)
Dear author,
You must to change the first map because you put in green just one island not all the country (Greece). This way the map has an scale error.
Best regards
Author Response
Please see the attachment.

This manuscript is a resubmission of an earlier submission. The following is a list of the peer review reports and author responses from that submission.
Round 1
Reviewer 1 Report
The research objective itself is current and necessary, even if it is not new.
The presented article does not have the character of a completed scientific study. If it should be a review article, the list of sources is too small. It gives the impression of a methodological concept with which the research itself should begin. The article does not show any data or findings.
The three intended case studies must show quantitative data and their qualitative interpretation. The presented tables are only preparation for carrying out the notified case studies.
In addition, the investigated parts of the three cities must be shown in the form of maps. It is necessary to show, for example, in Chania, whether the subject of research is a part of Splanzia or a larger historical part of the city. Findings cannot be understood without a graphic representation of the context. In addition to the clearly defined territory of the city, which is the object of the research, it is necessary to define the size of the statistical population samples with which the research was conducted. Sociological stratification is also worth mentioning. The conclusions presented in this form have general validity for all cities. Findings do not show any specifics for the Mediterranean region. E.G. the impact of gentrification is a global one.
The presented research basically does not provide any (new) results.
Reviewer 2 Report
Dear author,
After reviewing your work, I would like to inform you that it needs a greater degree of maturity, improvement in formatting (it is not adapted to the standards of the journal) and development.
I believe that the approach you intend to carry out is interesting but lacks a developed methodology, a deepening of the analysis you intend to cover and a broad corroboration or rejection of your initial hypothesis.
Nevertheless, please do not give it up. The case studies you present are interesting and could lead to results. Please try to use quantification techniques or categorise the variables you intend to analyse based on the literature. In addition, I encourage you to characterise the areas of study and justify them. Incorporate location mapping as well.
I recommend that you review the following works and cite them in the background:
https://ejtr.vumk.eu/index.php/about/article/view/2884
https://ejtr.vumk.eu/index.php/about/article/view/2627
Without further ado, much encouragement for the next phases.
